Effects of prolonged continuous computer gaming on physical and ocular symptoms and binocular vision functions in young healthy individuals

Lee Ji-Woo 1
Cho Hyun Gug 2
Moon Byeong-Yeon 2
Kim Sang-Yeob 2
Yu Dong-Sik yds@kangwon.ac.kr 2
1 Department of Optometry and Vision Science, Kyungwoon University , Gumi , South Korea
2 Department of Optometry, Kangwon National University , Samcheok , South Korea
Nock Nora
Electronic publication date: 2019 Jun 4
Publication date: 2019
Volume: 7
Electronic Location ID: e7050
Received 2018 Oct 11; Accepted 2019 May 1
Copyright: ©2019 Lee et al.
Copyright year: 2019
Copyright holder: Lee et al.
License: This is an open access article distributed under the terms of the Creative Commons Attribution License, which permits unrestricted use, distribution, reproduction and adaptation in any medium and for any purpose provided that it is properly attributed. For attribution, the original author(s), title, publication source (PeerJ) and either DOI or URL of the article must be cited.
License URL: https://creativecommons.org/licenses/by/4.0/

Keywords: Physical health, Computer gaming, Prolonged game, Ocular health, Visual functions

Funding: The authors received no funding for this work.

==============================
Background and Objective

Addiction to computer gaming has become a social problem in Korea and elsewhere, and it has been enlisted as a mental health disorder by the World Health Organization. Most studies related to computer use and vision have individually assessed physical and ocular symptoms and binocular vision. Accordingly, the present study comprehensively assessed subjective physical and ocular symptoms and functions related to binocular vision after prolonged continuous computer gaming. This study aimed to investigate the effects of prolonged continuous computer gaming on physical and ocular health and visual functions in young healthy individuals.

Methods

Fifty healthy college students (35 male/15 female), aged 19–35 years old, were enrolled in this study. The inclusion criteria were no binocular vision problems and no reported history of ocular disease. Participants played continuously for 4 h from 6:00 to 10:00 p.m. Physical and ocular symptoms and visual functions such as convergence, accommodation, phoria, and the blink rate were assessed before and after continuous computer gaming for 4 h.

Results

Continuous computer gaming for 4 h resulted in convergence and accommodation disturbances and increased physical and ocular discomfort. Near phoria showed an exophoric shift, whereas distance phoria showed no change. Moreover, the accommodative and vergence facilities and blink rate were significantly decreased. All visual functions recovered to the baseline levels by the following morning.

Discussion

Our findings suggest that excessive and continuous computer gaming impairs visual functions and causes ocular and physical fatigue. Our findings further the understanding of the adverse effects of excessive computer use on physical and ocular health, and adequate breaks are necessary to reduce physical and visual discomfort during computer gaming.

Introduction

The ability to visualize near objects clearly and accurately is necessary for individuals using computers and other digital screen devices on a daily basis. In today’s digital age, prolonged use of visual (or video) display terminals (VDTs) such as computers and smartphones has become very common and is one the main causes of ocular and physical discomfort (González-Pérez et al., 2014; Parihar et al., 2016). The majority of ocular and physical symptoms are associated with the effects of prolonged VDT usage on visual functions (Fenga et al., 2007; Tyrrell & Leibowitz, 1990; Yeow & Taylor, 1990; Yeow & Taylor, 1991). Computer vision syndrome (CVS), a condition associated with excessive computer use, is characterized by not only visual symptoms such as eyestrain, dry eyes, and blurred and double vision but also physical discomfort such as musculoskeletal shoulder disorders, neck pain, headache, and dizziness (Berqqvist et al., 1995; Hayes et al., 2007; Rosenfield, 2011). According to the Korea Games Users Survey Reports 2017, the rate of computer gaming in the general Korean population aged 10–65 years-old was increased from 67.9% in 2016 to 70.3% in 2017 (The Korea Creative Content Agency, 2017), which could lead to increased problems related to computer use.

Computer games have become a part of leisure activities in daily life, with young individuals spending excessive time indulging in such games (Wittek et al., 2016). The current most popular games require users to pay close attention to their VDTs, and the players show deep levels of emotional and physical addiction while gaming (Kim et al., 2016; Rikkers et al., 2016). Ubiquitous gaming cafes known as “PC bang” are very popular in Korea (Zastrow, 2017), where 24% of children are reportedly diagnosed with internet gaming addiction requiring hospitalization (Ahn, 2017). Computer gaming addiction can cause health and mental issues (Griffiths, Kussa & King, 2012; Saquib et al., 2017). Furthermore, excessive computer use has been reported to cause problems associated with poor computer (Toomingas, 2014). A recent study showed that the neck and shoulder are the most commonly reported areas affected by excessive computer use (James et al., 2018). Other commonly reported symptoms include headache, eyestrain, double vision, dry eyes, and ocular fatigue (Akinbinu & Mashalla, 2014). Some studies (Qu et al., 2005; Yekta, Pickwell & Jenkins, 1989) have also reported that near work on VDTs may cause not only ocular and physical symptoms but also transient changes in refraction, visual acuity, accommodation, and convergence. Tosha et al. (2009) examined the relationship between visual discomfort and the accommodation response in a college population with visual discomfort and found that increased visual discomfort is characterized by accommodative fatigue, with a higher lag of accommodation developing at a near viewing distance over time.

Furthermore, a study of over 1,173 Korean school-going children reported that the participants spent approximately 1.35 h on computer-based academics and 2.03 h on computer gaming during a normal day, and computer-using time was 3.6  ± 2.2 h for internet-addicted Korean adolescents as reported by Yang et al. (2014). Computer overuse has led to progressive negative effects yearly in the users’ physical and mental health, and the “shutdown law” was recently introduced in Korea for the protection of youth from internet-based computer games (Kim et al., 2016; Sung, 2014). The World Health Organization (WHO, 1987) has recommended that frequent minibreaks (of few seconds duration) must be taken while working with VDTs to prevent problems, while longer breaks are often advisable after one or two hours—depending on the job. However, it is difficult to ensure breaks in individuals who are engrossed in computer games. Actually, most individuals addicted or immersed to computer games tend to play for prolonged hours without a break, and the effects of prolonged continuous viewing of VDTs at a near distance on physical and ocular health remain unclear. Moreover, most studies related to computer use and vision has individually evaluated physical and ocular symptoms and binocular vision (Parihar et al., 2016; Hamed, David & Marzieh, 2013). Unlike many studies reported in the literature related to VDT, the current study examined physical and ocular symptoms and signs (binocular functions) resulting due to prolonged computer gaming without rest for 4 h (hours) at a time, which has not been evaluated till date. Therefore, we believe that a study focused on a comprehensive assessment of subjective physical and ocular symptoms and functions related to binocular vision after prolonged continuous computer use is necessary. Accordingly, the aim of this study was to investigate subjective physical and ocular symptoms and visual functions, as well as their interrelationships, after prolonged continuous computer gaming in young healthy individuals.

Methods

Participants

This research complied with the tenets of the Declaration of Helsinki and was approved by the Institutional Review Board of Kangwon National University (KWNUIRB-2017-07-002-003). Informed consent was obtained from each participant.

Minimal required sample size was performed using GPower 3.1.2 software (Heinrich-Heine-Universität, Düsseldorf, Germany). With an alpha error of 0.05, 90% power, effect size of 0.25, 1 group, and 3 measurements, a sample size of 43 was calculated but fifty Korean healthy college students (age, 19–35 years, average age, 22.5 (mean)  ± 3.4 (SD, standard deviation) years; male:female, 35:15) who had maintained a daily routine without physical and ocular fatigue in their daily life were randomly enrolled as subjects for our study. The inclusion criteria were as follows: no near or distance vision problems; a corrected or uncorrected distance visual acuity of 20/25 (0.8) or better in each eye; normal stereopsis; and no reported history of ocular disease. Twenty-nine subjects (58%) wore spectacles, and their mean spherical equivalent power was −1.99  ± 2.53 diopters (D).

This study was performed under immersive computer gaming that involved general tasks for a prolonged period (4 h) without a break versus a short period (≤2 h) of VDT exposure in order to determine the presence of physical and ocular discomfort.

Procedures

The subjects were asked to play a computer game (Diablo III; Blizzard Entertainment, Irvine, CA, USA) on a desktop computer screen (CX501N-KN/KOR with a 15-inch LCD monitor, Samsung, Korea) from a viewing distance of 50 cm under room illumination (approximately 50 lux). They played continuously for 4 h from 6:00 to 10:00 p.m. All subjects received a modified questionnaire (Ames, Wolffsohn & Mcbrien, 2005) designed for assessing the effects of virtual reality viewing on monocular, binocular, and physical symptoms, as well as ophthalmological examinations for the assessment of visual functions, before and after the gaming session.

The questionnaire included 13 items pertaining to four physical and nine ocular symptoms. Each item was graded on a scale of 0–4 (0 = none, 1 = slight, 2 = moderate, 3 = severe, and 4 = very severe).

Visual functions were measured under habitual viewing conditions. Visual acuity was measured using visual charts (ACP-8; Topcon, Tokyo, Japan) and a phoropter (CV-3000; Topcon, Tokyo, Japan). The near point of convergence (NPC) is to evaluate the convergence amplitude, which represents a visual function to obtain single vision, was measured by bringing an accommodative target to the nose and recording the point time at which the subject could see double (Scheiman et al., 2003). The near point of accommodation (NPA) is the near point to measure the amplitude of accommodation under binocular conditions, which represents a visual function for maintaining a clear image (Abraham et al., 2005), was measured using an accommodative convergence rule (GR50; Bernell, Mishawaka, IN, USA) while the subject attempted to read small letters (near visual acuity, 20/32). Phoria is to assess the presence, direction, and amplitude of the eye alignment, which indicates latent misalignment of the eyes, was measured using Howell phoria cards (Wong, Fricke & Dinardo, 2002) (CDHP; Bernell, Mishawaka, IN, USA) at 3 m and 33 cm. Negative values indicate exophoria, whereas positive values indicate esophoria. Accommodative and vergence facilities is to evaluate the ability of the accommodative response and ability of the fusional veregence respectively (Weissberg, 2004; Gall, Wick & Bedell, 1998), which were measured using a ±2.00 D binocular flipper lens and a prism flipper (3Δ base-in + 12Δ base-out; Δ: prism diopter) at 40 cm. Blink rate was measured using a camcorder combining a video camera (NV-GS400; Panasonic, Kadoma, Japan), complete and incomplete blinks were distinguished by the presence of the full and partial eye cycle of open-close-open. Binocular function was measured at time-points of before and after the computer game, and the next morning.

Statistical analysis

All data were statistically analyzed using SPSS (ver. 18.0 for Windows; SPSS Inc., Chicago, IL, USA). Pearson’s correlation analysis, paired t-tests, and repeated measures analysis of variance (ANOVA) were used for statistical analyses. The level of significance was set at p = 0.05. If significant differences were found, post hoc tests with Bonferroni corrections were used to identify the level of significance. Partial missing values in phoria and blink rate were excluded from descriptive statistics analyses.

Results

The mean scores for the four physical (monocular) and nine ocular symptoms associated with computer game use are shown in Table 1. The mean overall scores for the physical and ocular discomfort domains increased from 0.39  ± 0.78 and 0.50  ± 0.83, respectively, before the gaming session to 1.61  ± 1.24 (t = 15.7, p < 0.001) and 1.40  ± 1.22 (t = 17.44, p < 0.001), respectively, after the gaming session. The scores for all individual items also showed significant changes after the gaming session, and the mean difference was 1.00  ± 0.33. In the physical discomfort domain, the score for neck discomfort after the gaming session was the highest, followed by the scores for shoulder discomfort, headache, and back discomfort. In the ocular discomfort domain, the score for tired eyes was the highest (2.30  ± 1.09), followed by the scores for dry eyes, blurred vision, and eyestrain. The score for itchy eyes was the lowest (0.82  ± 1.00).

Table 1 Changes in physical and ocular symptom scores after continuous computer gaming for 4 h.

Symptoms	Score, mean  ± SD	Significance*	
	Before gaming	After gaming		
Physical discomfort	0.39  ± 0.78	1.61  ± 1.24	t = 15.7, p < 0.001	
Shoulder	0.48  ± 0.81	1.88  ± 1.19	t = 8.80, p < 0.001	
Neck	0.44  ± 0.81	1.92  ± 1.21	t = 9.78, p < 0.001	
Back	0.20  ± 0.53	1.16  ± 1.15	t = 6.60, p < 0.001	
Headache	0.42  ± 0.91	1.46  ± 1.30	t = 6.76, p < 0.001	
Ocular discomfort	0.50  ± 0.83	1.40  ± 1.22	t = 17.44, p < 0.001	
Red eyes	0.48  ± 0.81	1.38  ± 1.09	t = 7.18, p < 0.001	
Eyestrain	0.44  ± 0.79	1.58  ± 1.03	t = 8.32, p < 0.001	
Itchy eyes	0.44  ± 0.79	0.82  ± 1.00	t = 2.57, p = 0.013	
Tired eyes	0.94  ± 0.98	2.30  ± 1.09	t = 8.73, p < 0.001	
Dry eyes	0.64  ± 1.01	1.64  ± 1.34	t = 6.86, p < 0.001	
Teary eyes	0.34  ± 0.80	0.88  ± 1.14	t = 3.62, p < 0.001	
Irritated eyes	0.34  ± 0.66	0.96  ± 1.12	t = 3.73, p < 0.001	
Blurred vision	0.46  ± 0.73	1.60  ± 1.20	t = 7.28, p < 0.001	
Aching eyes	0.40  ± 0.76	1.40  ± 1.29	t = 6.29, p < 0.001	
Notes.

Symptoms were scored on a scale from 0 (none) to 4 (very severe).

SD standard deviation

* The p-values were determined using paired t-tests. Differences between the means were statistically significant (p < 0.05).

The effects of prolonged continuous computer gaming on visual functions are shown in Table 2. The mean NPC values showed significant changes immediately after and the morning after the gaming session (F = 47.83, p < 0.001). A post hoc Bonferroni multiple comparisons test indicated that NPC significantly increased immediately after the gaming session (post hoc p < 0.001 for before and after) and recovered to the baseline level by morning (post hoc p = 1.000 for morning and before). The mean NPA value also showed significant changes (F = 42.50, p < 0.001). A multiple comparisons test with Bonferroni correction showed that the NPA value after the gaming session was significantly different from the next-day value and the baseline value (host hoc p < 0.001 for before and after, p < 0.001 for after and morning); the next-day value and the baseline value showed no significant differences (post hoc p = 1.000 for morning and before).

Table 2 Changes in binocular function after continuous computer gaming for 4 h.

Binocular function (Expected values)a	Before gaming (a)	After gaming (b)	Following morning (c)	Significance (Post hoc)d	
NPC (cm) (2.5  ± 2.5)	7.23  ± 1.64	8.77  ± 1.60	7.36  ± 1.29	F = 47.83, p < 0.001(b > a, c)	
NPA (cm) (NA)	7.79  ± 1.44	9.11  ± 1.93	7.78  ± 1.49	F = 42.50, p < 0.001(b > a, c)	
Phoria at distance (Δ) (1  ± 1 exophoria)	−0.76  ± 1.71	−0.83  ± 2.25	−0.62  ± 1.69	F = 1.67, p = 0.193 (ns)	
Phoria at near (Δ) (3  ± 3 exophoria)	−3.73  ± 3.93	−5.75  ± 4.85	−3.68  ± 4.86	F = 18.39, p < 0.001(b > a, c)	
Accommodative facility (cpm) (10  ± 5)	14.42  ± 3.20	12.68  ± 3.98	15.54  ± 2.95	F = 30.08, p < 0.001(c > a > b)	
Vergence facility (cpm) (15  ± 3)	16.94  ± 3.16	15.02  ± 3.67	17.70  ± 3.41	F = 38.97, p < 0.001(c > a > b)	
Blink rate (per minute) (12–19)	16.24  ± 5.14	(8.27  ± 5.40)b	(9.51  ± 5.28)c	F = 106.29, p < 0.001(a > b, c)	
Notes.

Data are expressed as means  ± standard deviations.

NPC near point of convergence

NPA near point of accommodation

NA not applicable due to the value determined by age or refractive errors

Δ prism diopter

cpm cycles per minute

ns non-significant

a Expected values from Scheiman & Wick (2002), Weissberg (2004) and Karson et al. (1981).

b Blink rate after 1 h.

c Blink rate after 4 h.

d Bonferroni correction for multiple comparisons.

The p-values were determined using repeated measures analysis of variance. Negative and positive values for phoria represent exophoria and esophoria (horizontal), respectively. Missing data are 15 persons for phoria and 10 persons for blink rate.

The accommodative facility (AF) and vergence facility (VF) significantly decreased after the gaming session (post hoc p < 0.001 for before and after in both facilities). Both facilities recovered to more than baseline levels by the following morning (post hoc p = 0.001 and p = 0.045 for morning and before in AF and VF, respectively).

The blink rate before the gaming session was 16.24 times per minute; this significantly decreased to 8.27 and 9.51 times per minute at 1 and 4 h after the start of the game, respectively (F = 106.29, p < 0.001; post hoc p < 0.001 for before and after, p = 0.080 for after 1 h and 4 h).

The changes in phoria during the computer gaming session are shown in Fig. 1. Near phoria showed an exophoric shift (exophoria) during the session (repeated measures ANOVA; F = 24.25, p < 0.001, post hoc p = 1.000 for after 1 h and 2 h, p = 0.010 for after 2 h and 3 h, p < 0.001 for after 3 h and 4 h) and recovered to baseline levels by the following morning (post hoc p = 1.000 for morning and before). Distance phoria showed no significant differences during the game (repeated measures ANOVA; F = 0.61, p = 0.610). Horizontal phoria at distance showed no statistically significant differences among before, after, and the day after the gaming session (F = 1.67, p = 0.193), whereas horizontal phoria at near changed from −3.73  ± 3.93 before the gaming session to −5.75  ± 4.85 and −3.68  ± 4.86 immediately after and the day after the gaming session, respectively (F = 18.39, p < 0.001; post hoc p < 0.001 for before and after, p < 0.001 for after and morning, p = 1.000 for morning and before).

Figure 1 Changes in phoria during a continuous 4-h computer gaming session.

The symbols and error bars represent mean and standard deviations, respectively. Negative values denote exophoria and positive values denote esophoria (horizontal).

Pearson’s correlation analysis showed a very weak negative correlation between itchy eyes and NPC after the gaming session (r =  − 0.294, p = 0.039) and a weak positive correlation between tired eyes and NPA (r = 0.361, p = 0.01) and between blurred vision and NPA (r = 0.298, p = 0.036) after the gaming session.

Discussion

Our results are not limited to gaming activities, but applicable under the environment of overuse of personal computers for the purpose of computer game after the college students’ work day.

There were several differences between the current study and the previous studies (Yeow & Taylor, 1990; Yeow & Taylor, 1991; Qu et al., 2005). Our study describes game-based VDT activity in contrast to work-based VDT, continuous 4-h exposure in contrast to short-term 1–2.35-h or long-term exposure, physical and ocular symptoms in contrast to ocular symptoms alone, assessment of accommodative facility and vergence facility besides NPC, NPA, phoria as binocular functions, and measured sessions of a recovery point besides before and after. In the present study, we found that prolonged continuous computer use for gaming resulted in both physical and ocular discomfort as well as changes in binocular functions. In particular, the neck and shoulder, which remain in the same posture while playing, were affected. The major visual symptom was ocular fatigue (tired eyes), followed by dryness and blurred vision. These findings are consistent with those of other studies (Hayes et al., 2007; Klussmann et al., 2008) reporting a high prevalence of neck and shoulder symptoms, ocular fatigue, and blurred vision after excessive computer use. The findings of a 13-item questionnaire used in our study revealed higher scores for physical symptoms than for ocular symptoms after 4 continuous hours of gaming, although tired eyes was associated with the highest score. These results indicated that the physical problems related to the neck, shoulders, and backs were the most affected regions among the computer users, other ocular symptoms were the most frequently occurring health problems (Lanhers et al., 2016), and of the 59 included workers who use a computer for 3–6 h per day, the musculoskeletal problems were reported by 71.1% of subjects as compared to that for visual problems of 61.0% out (Talwar et al., 2009); thus, after physical discomfort, discomfort of the eye is the second most frequent problem reported by VDT operators. Tired eyes can be explained by the fact that high accommodation and convergence values are required for maintaining a clear vision at small distances from VDTs, and these values need to be maintained for the entire gaming duration. This invariably results in ocular fatigue. On the other hand, physical factors associated with discomfort had lower scores than did tired eyes due to frequent changes in the body posture to relieve physical stress.

We also found changes in visual functions after continuous computer gaming for 4 h in the present study. NPC, a visual function used to achieve single vision during near-distance work, showed a significant increase after the gaming session and recovered to the baseline level the next day. Qu et al. (2005) found that NPC was significantly increased after short-term VDT use. However, Yeow & Taylor (1991) found that NPC decreased with an increase in age, with no significant difference between VDT and non-VDT users. Hamed, David & Marzieh (2013) reported that the evaluation of NPC helped in the differentiation of symptomatic and asymptomatic subjects, and that the average NPC value for break point was 11.7 cm. While NPC is a measurement of how close one can bring a fixation target to the nose while maintaining fusion, NPA is used to assess the amplitude of accommodation. Accommodation is connected with the function of vergence maintained by extracocular muscles. Because of the interaction that occur between accommodation and vergence, the accommodation and convergence disturbances observed after the gaming session in the present study can be attributed to intra- and extra ocular muscle fatigue resulting from the prolonged continuous computer gaming.

Accommodative and vergence facilities permit the ability to sustain clear and single binocular vision during near-distance work. A healthy individual’s accommodative facility should be binocularly clear within 10.0  ± 5 cpm (Scheiman & Wick, 2002). In the present study, the value showed a significant change from 12.68 to 15.54 cpm. The vergence facility for single binocular vision also showed a significant change from 15.02 to 17.70 cpm; the norm is 15 cpm (Weissberg, 2004). These changes may be associated with symptoms such as ocular fatigue, eyestrain, and blurred vision, although NPC and NPA showed significant changes within normal values. Both accommodative and vergence facilities recovered by the next morning.

The results of the present study also showed that near phoria (dissociated phoria) increased from 3.73Δ exophoria to 5.75Δ exophoria with the development of symptoms after continuous computer gaming for 4 h, and it subsequently recovered to 3.68Δ exophoria by the following morning. This result is consistent with those of other studies (Yekta, Pickwell & Jenkins, 1989; Tsubota & Nakamori, 1993) showing an exophoric shift in phoria due to near-distance tasks, with no significant change in distance phoria over time. According to other studies related to changes of phoria, Pickwell, Jenkins & Yekta (1987) reported that 30 min of reading in inadequate illumination or from an abnormally close distance resulted in an increase in phoria with the development of ocular symptoms. Also, Yekta, Pickwell & Jenkins (1989) found that near-dissociated phoria increased from 4.70Δ exophoria to 5.20Δ exophoria after 30 min of reading and decreased to 4.87Δ exophoria after 30 min of relaxation, with no accompanying symptoms. Collectively, these findings including our results suggest that near phoria for binocular vision changes to exophoria under the effect of visual stress caused by factors such as inappropriate illumination, abnormal working distance, and prolonged working hours.

The mean blinking rate at rest reportedly varies from 12 to 19 blinks per minute (Karson et al., 1981). Tsubota & Nakamori (1993) observed 22 blinks per minute under relaxed conditions and seven blinks per minute during VDT work. Patel et al. (1991) reported 18.4 blinks per minute before a computer task and 3.6 blinks per minute during the computer task. We found that the number of blinks per minute significantly decreased from 16.24 before the gaming session to 8.27 after 1 h of gaming and 9.51 after 4 h of continuous gaming. However, we do not know whether the decreased blink rate induces ocular fatigue, or vice versa. However, the blink rate may be an effective index for assessing visual fatigue during VDT tasks. Intentional blinking during VDT tasks is required to minimize visual fatigue. Blinking exercises have been shown to improve the glandular function and reduce the frequency of incomplete blinks (Murakami, Blackie & Korb, 2014; Downie & Craig, 2017). Therefore, we highly recommend blink training to increase the blinking rate during prolonged computer use.

Pearson’s correlation analysis showed that ocular fatigue and blurred vision were weakly related to NPA, whereas itchiness was weakly related to NPC. These subjective symptoms may be provoked by the accommodation and convergence disturbances induced during prolonged computer gaming. Although these changes are usually temporary, visual fatigue caused by long-term VDT use tends to accumulate over time as reported by Murata et al. (1996).

Our study has a limitation that the results have not been compared with those of a control group. However, Qu et al. (2005) compared VDT and non-VDT users and reported that even short duration (1 h) of use of VDT leads to reduction in NPA, receding of NPC, and increase in near lateral exophoria. It is evident that signs derived from computer operation involves near work. VDT exposure in our study has more accelerated conditions (highly stressful conditions) than in the study by Qu et al. Therefore, we are of the opinion that the findings of the current study pertaining to physical and visual discomfort and binocular functions following continuous computer gaming are significant even in the absence of a control group. This study is limited by the small sample size that included participants with healthy eyes without binocular vision problems so that compounding variables associated with physical and ocular fatigue in the daily life of the individual were minimized. A larger sample of participants with binocular disorders is necessary for extensive estimation of the effects of prolonged continuous computer gaming. Also, the appeared points of physiological effects and the recovery were not exactly examined in this study.

The time-points of physiological effects and recovery were not exactly examined in this study. Nevertheless, our findings offer valuable insights and provide direct evidence of significant changes in physical and ocular symptoms and visual function after a 4-h period of computer game. The time-points of physiological effects could be variable due to influencing factors such as individual and ergonomic factors. The decreased accommodative functions in the subjects after performing VDT task for 2 h were recovered at the end of the 1-h lunch break (Saito et al., 1994). Accommodative functions after resting for 0.5 h following 1.5 h VDT task showed a tendency to return to the previous normal value (Yoo, Yoon & Kim, 1992). These previous studies suggest that the recovery time of symptoms and signs may be depended on the strength of VDT work.

Conclusions

In summary, the findings clearly indicate that changes in visual functions evoke ocular symptoms after prolonged continuous computer gaming, which also induces physical symptoms. Although these symptoms are usually temporary, the accumulation of fatigues could be potential risk factors for irreversible physical and visual disturbances associated with computer overuse. Users may need rest to alleviate fatigue due to computer use. Moreover, individuals should be advised regarding their working hours. Therefore, it seems reasonable that all individuals should be advised to take breaks and periodically gaze into the distance to minimize accommodation and vergence disturbances; longer rest periods will result in lesser physical and ocular fatigue.

Supplemental Information

File S1 Raw data for data analyses and Tables 1, 2 and Fig. 1

Click here for additional data file.

Supplemental Information 1 Questionnaire form

Click here for additional data file.

Additional Information and Declarations

Competing Interests

Author Contributions

Human Ethics

Data Availability

The authors declare there are no competing interests.

Ji-Woo Lee and Dong-Sik Yu conceived and designed the experiments, performed the experiments, analyzed the data, contributed reagents/materials/analysis tools, prepared figures and/or tables, authored or reviewed drafts of the paper, approved the final draft.

Hyun Gug Cho conceived and designed the experiments, analyzed the data, contributed reagents/materials/analysis tools, prepared figures and/or tables, authored or reviewed drafts of the paper, approved the final draft.

Byeong-Yeon Moon and Sang-Yeob Kim conceived and designed the experiments, analyzed the data, contributed reagents/materials/analysis tools, wrote the paper.

The following information was supplied relating to ethical approvals (i.e., approving body and any reference numbers):

This research complied with the tenets of the Declaration of Helsinki and was approved by the Institutional Review Board of Kangwon National University (KWNUIRB-2017-07-002-003).

The following information was supplied regarding data availability:

The raw measurements are available in a Supplemental File.

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
