# Peer review of "Effects of prolonged continuous computer gaming on physical and ocular symptoms and binocular vision functions in young healthy individuals"

_PeerJ, doi:10.7717/peerj.7050_

## Round 0.1 · original submission · Major Revisions

Your manuscript has been reviewed and found to need major revisions before further consideration. If you choose to resubmit, please provide a point-by-point response on how each issue was addressed in the revised manuscript.

·

Basic reporting

The quality of technical writing of the manuscript is excellent, and the prose is easy to read. Relevant previous literature is cited. There are some important omissions, noted below, which appear to have tackled a very similar question.

Minor issues:

line 61. 24 % *of* children

line 185. Given that body posture can be frequently changed during game play, it is unclear why higher scores should be obtained for physical symptoms than ocular symptoms (apart from tired eyes).

Experimental design

The research question, and experimental methods employed to investigate it, are both well defined. However, as discussed in further detail below, it has not been convincingly demonstrated that this research fills an identified knowledge gap. The authors should clarify how their study provides a greater understanding of the effects of short- or prolonged computer gameplay on visual function, as differentiated from that of VDT exposure that does not involve computer gaming.

Validity of the findings

The results reported appear to be reasonable, and the comparisons to similar results from related studies were helpful to put the findings in context of previous literature. There are several issues, however, that should be addressed to make clear the contribution of the findings:

1. Surprisingly, the authors have not addressed what would seem to be the most obvious question, regarding whether there is a difference in the effects of prolonged VDT exposure from non-gaming use, e.g., of programmers. Are the observed effects in fact due to gameplay activity or do they manifest equally in any interactive (or perhaps even passive) screen-viewing activity?

It is well established that VDT use affects visual function. As one example, cited by the authors, (Qu XM, Chu RY, Wang L, Yao PJ, Liu JR. 2005. Effects of short-term VDT usage on visual functions. Zhonghua Yan Ke Za Zhi 41(11):986–989 DOI: 10.3760/j:issn:0412-4081.2005.11.007) conclude that ``Short-term VDT work does have a significantly greater temporarily [sic] effect on visual function, tear film quality and visual quality.'' However, as described by other literature, similar effects are observed in other (non-VDT) near field work. Studies by Yeow and Taylor report that ``there are no significant differences in type, number and frequency of the work-related symptoms between VDT users and non-VDT users'' (P.T. Yeow, S.P. Taylor, The effects of long-term VDT usage on the nature and incidence of asthenopic symptoms, Applied Ergonomics, Volume 21, Issue 4, 1990, https://doi.org/10.1016/0003-6870(90)90199-8) and ``Results show that VDT work does not have a significantly greater effect on visual function than non-VDT work.'' (Yeow and Taylor, Effects of short-term VDT usage on visual functions, Optom Vis Sci. 1989 Jul;66(7):459-66).

2. Beyond the direct reporting of the experimental results, a meaningful discussion of the implications of the data would have strengthened the manuscript. Instead, the Conclusions state that the ``visual fatigue caused by long-term VDT use may tend to accumulate over time.'' The duration of the experiment was, of course, limited to a single 4-hour game session, and thus, this hypothesis is offered without substantiation or supporting citations.

3. Apart from blink rate, for which data were reported only before and during gameplay, all of the experimental measures returned to baseline values by the following morning. The authors should therefore elaborate on the contention that their study provides ``the first direct evidence of changes in physical and ocular health and visual functions after prolonged continuous computer gaming''. One might argue, to the contrary, that these functions manifested changes only during and immediately following gameplay, when such physiological effects are entirely anticipated, just as one's heart rate and respiration increase during periods of physical activity. The more important question, I would argue, is recovery time: do these effects persist beyond a typical non-VDT-centric-activity ocular recovery time following cessation of the activity? If not, this warrants comment.

Reviewer 2 ·

Basic reporting

1. On table 1, authors should identify between each group the differences were found statistically significant.
2. The meaning of the negative and positive values for phoria should be in the text.

Experimental design

1. The major drawback of the study is the lack of a control group.
1.Why VA was measured with the phoropter? This is not a normal vieweing condition.
2.The PPA was measured monocularly or binocularly
3. Why all the accommodation parameters were measures binocularly? Any change in binocular vision will have impact in the binocular accommodation parameters even if the accommodation will not suffer of any change.
4. In the methods it is not explained that measurements were repeated in the next morning.

Validity of the findings

The authors state that " Although these changes are usually temporary, visual fatigue caused by long-term VDT use may tend to accumulate over time". However, this can not be concluded by this study.
They also say that "ergonomically designed work spaces are desirable to minimize physical discomfort after long VDT tasks" but no study was done to evaluate it.
Authors should rewrite their conclusions.

Reviewer 3 ·

Basic reporting

See comments

Experimental design

See comments

Validity of the findings

See comments

Additional comments

Review of the manuscript: “Effects of prolonged continuous computer gaming on physical and ocular health and visual functions in young healthy individuals”

The manuscript reports results of a study investigating effects of 4 hours of continuous computer gaming (“the task”) in certain aspects of the visual system. The study shows that near phoria changed after the task and that was also accompanied by an increased score in serf-reported visual discomfort and changes in blinking rate. The results of the study are interesting and likely to attract the attention of the scientific and clinical community. The presentation of the findings can be improved to help readers that may be unfamiliar with some of the parameters measured. It is unclear from the text which was/were the hypothesis or hypotheses. During the discussion and conclusions, there is a tendency to go beyond what the results show or can prove. I have a few suggestions that will significantly improve the manuscript.

# Title
The title is misleading and should be changed. The study is not investigating any health or disease issues but instead is investigating changes in self-reported discomfort – which does not correspond necessarily to any unhealthy state. Visual function is also misleading because, for example, visual acuity is not part of the main results. I suggest replacing it by “binocular vision function”. A good example of the phrasing is what is written in line 82: …” … computer gaming on physical and ocular symptoms and binocular vision in young …” which is what was actually done in the study.

# Introduction
The text is full of vague statements that should be removed or replaced by concrete contents, some examples:
line 43 - “In today’s generation”, what is today’s generation?

Line 52-54 – it seem like authors are trying to suggest that people should stop using computers, maybe this sentences should be re-written;

line 62 – “Computer gaming addiction can cause health and mental issues”, what is the relationship of this sentence with poor visual ergonomics? Ergonomics seem to be the topic of the paragraph (not mental health);

line 77 – “Computer overuse has been deteriorating the visual environment year after year”, please specify the meaning of “deteriorating the visual environment”, “Actually, most individuals addicted to computer games”, What defines an addicted person?

Line 82 – “Moreover, most studies related to computer use and vision have individually evaluated physical and ocular symptoms and binocular vision.”, if this is the main justification for this study, this evidence should be given in the introduction. As it is this statement comes from nowhere;

# Methods
Line 95 – “Sample size was estimated based on the difference of computer-using time between a 3.6 ± 96 2.2 hours per day and a 2.4 ± 1.6 hours per day for internet-addicted and non-addicted Korea adolescents in the previous study.” Is not clear why and how does the sample size comparing addicted and non-addicted user has been used if there is only one group in the study.

Line 109 – an English version of the questionnaire should be included in the extra material and a clear reference should be given;

Line 116 – 128 – here there are different tests for different “visual functions”. I recommend that authors define what each function mean before explaining the measurement. For example, accommodative and vergence facility are two variables only defined in discussion. Also, a brief explanation why these measurements are important should be included, here or in introduction. In general, the introduction is very vague about which vision problems can arise because of gaming;

Line 128 – the words “finally, the” should be removed from the text and the method of counting explained. I assume that the camera does not count blinks (complete / incomplete blinks should be defined);

# Results
The statistical tests used seem adequate. Although, given the amount of results presented, normative values for each test presented in the same table as the experimental data (Table 2) would improve the readability of the results/manuscript;

Is not clear what do the plus or minus sign means throughout all the results -- that should be described;

Line 163-174, I assume all results comparing “before-after-day after” mentioned here are included in Figure 1, therefore this figure should be mentioned before reporting the findings;

Also, lines 163-166 are about anova results and, I presume, lines 171-174 still anova findings. Describe the anova model, e.g. which factors were used. The correlation results in between anaova outcomes should be removed and all anova findings reported together. The text saying “ no change” should be reviewed. Authors should be aware that there were changes, but they weren’t statistically significant – please read more about this here: https://www.ncbi.nlm.nih.gov/pubmed/7647644. In addition, for example in line 164 “whereas horizontal phoria at near changed from −3.73 ± 3.93”, which test has been used? p-value? The same for other significant results;

What is the different between these results? And what is the difference between “shift” and “change”: Lines 173-174: “Distance phoria showed no shift during the game (repeated measures ANOVA; F = 0.61, p = 0.610)”. and Line163: “Horizontal phoria at distance showed no change after the gaming session (F = 1.67, p = 0.193)”;

# Discussion
Line 178: Can authors please explain in the manuscript the meaning of “ impaired visual functions”? If they don’t have a simple explanation, maybe should consider re-writing.

Lines 200-202: Is unclear for me why authors bring NPA in line 199 and then conclude that their findings are related with intra and extraocular muscles fatigue, a better connection between the findings and the causes should be given;

Paragraph line 203 - 233, discusses poor illumination but authors do not explain how that applies to their study. Authors should relate the discussion with illumination in the room where the experiment took part and/or the luminance of the screen. This paragraph needs further clarification and connection between previous findings and the findings of the present study;

Lines 229-233, ideas are very foggy here and the final recommendation is unsupported by the findings. This part of the discussion needs further work and authors should be cautious about the recommendation they do based on their results;

Line 238-243: authors need to explain why they consider that the sample was small when they use more subjects that the sample size that they computed and described in methods. Also, I cannot understand which “physical and ocular health” aspects changed during the study. Why do further studies need to include people with binocular disorders is also unclear to me. It seem that this single paragraph puts together the limitations and the main findings of the study, which is somewhat contradicting.

Lines 252-257, It is unclear for me as reviewer what support does this study provides to the set of recommendations given. Unless authors can clarify what they mean what why these lines should be part of the manuscript, I recommend them to be removed.

---

## Round 0.2 · Major Revisions

The revised manuscript submitted has not addressed many concerns raised in the initial review by multiple reviewers. The authors need to fully address all issues not just in the rebuttal letter but in the actual manuscript. If the authors are willing to do so, please submit a revised manuscript and revised rebuttal letter that details each issue raised and how this was addressed in the revised manuscript.

·

Basic reporting

See below

Experimental design

See below

Validity of the findings

See below

Additional comments

I remain uncertain as what contribution the present work offers to the literature. The table (R.1) provided in rebuttal provides a succinct comparison of the author's approach and findings with that of prior literature cited. However, neither this table, nor the following text that elaborates on this body of work, establishes an unexpected difference in ocular discomfort ("temporary effect on visual function") from the "accelerated" (4 hours without break) exposure to game-based VDT activity vs. that of Qu et al.'s short-term (1 hour) VDT exposure. The possible exception is that of physical (non-ocular) discomfort, which was, I would guess, not investigated by Qu et al. This problem is exacerbated by the lack of a comparison group, e.g., non-gaming VDT users, exposed to a display for similar periods of time, that would otherwise plausibly offer an opportunity to explore such a comparison. Since the present manuscript does not focus predominantly on physical effects, the question remains: "what does a reader learn about the impact of VDT exposure (whether for one hour or four hours; whether for gaming or non-gaming activities) that is not already known from the literature?"

With regard to the point I raised in my initial review, the claim that the study "provide[s] the first direct evidence of changes in physical and ocular health and visual functions after prolonged continuous computer gaming" seems possibly hyperbolic, given the past evidence for, at the very least, such ocular effects from extended VDT exposure. Moreover, these results would seem, intuitively, consistent with expectations for any activity demanding long-term visual attention in a seated position. However, I appreciate the addition of the references to visual function recovery time in other studies, as well as the addition of Murata et al. (1996) as a citation to accumulation of visual fatigue over time.

Reviewer 2 ·

Basic reporting

This reviwed version is a much better version of the work.

Experimental design

Although authors have answer to all my concerns, I still have some issues regarding this part of the work

1. On table 2, authors should identify between each group the differences were found statistically significant.

Validity of the findings

N.A.

Additional comments

This reviwed version is a much better version of the work.

Reviewer 3 ·

Basic reporting

(1.1) In my previous review I asked the authors to add the expected values for variables in Table 2, and authors answered "normative values for each test could not be determined". This is a misinterpretation of my request, expected values for binocular and accommodative functions exist and this reference https://www.ncbi.nlm.nih.gov/pubmed/28095087 is an example. I still think expected values for binocular and accommodative functions should be in Table 2 to certify that participants in the current study showed baseline values within the expected range.


(1.2) line 98 in all the manuscript - what is the meaning of "± 2.2" is 2.2 the standard error?, standard deviation of the mean? the inter-quartile range? please specify before using.

(1.3) This statement would fit better in discussion: "Our results are not limited to gaming activities, but applicable under the environment of overuse of personal computers for the purpose of computer game after the college students’ work day."

(1.4) Blinking rates should be presented with a central tendency measure (e.g. mean) and a measure dispersion (e.g. standard deviation);

(1.5) Figure 1, what do the symbols represent? mean values?

(1.6) Lines 163-184, it would be easier to understand the results if mean differences for statistically significant changes are reported;

Experimental design

(2.1) Sample size calculation is unclear, authors use hours-of-computer-use (HCP) obtained from addicted and non-addicted internet users from a previous study to determine the sample size needed to test their hypothesis. In the current study HCP are fixed (not an outcome mesure); therefore, I cannot understand why authors use such a variable (HCP) to determine the sample size. Also, the hypothesis of the study should be more clear if authors want readers to understand their sample size calculations.

(2.2) How can participants can be randomly enrolled when there is only one group? please clarify

(2.3) Given (2.2) I am unable to understand the limitations described in lines 265-268

Validity of the findings

'no comment'

Additional comments

GC1: authors should verify if "divergent shift" is the correct term to express the observed changes in phorias;

GC2: The last sentence of the abstract starting "and highlight the need for ..." is not supported by the current study and author should delete it;

GC3: Reference Ahn appears with years 2017 in the text and 2007 in the reference list, please correct;

GC4: unless authors can defend the use of the expression "deterioration of the visual environment" this expression should be replaced by a more objective one;

---

## Round 0.3 · accepted · Accept

The revised manuscript has been reviewed and found acceptable.

Reviewer 2 ·

Basic reporting

NA

Experimental design

NA

Validity of the findings

NA

Additional comments

NA